# Conserved Double Translation Initiation Site for Δ160p53 Protein Hints at Isoform’s Key Role in Mammalian Physiology

**DOI:** 10.3390/ijms232415844

**Published:** 2022-12-13

**Authors:** Maria José López-Iniesta, Shrutee N. Parkar, Ana Catarina Ramalho, Rafaela Lacerda, Inês F. Costa, Jingyuan Zhao, Luísa Romão, Marco M. Candeias

**Affiliations:** 1MaRCU—Molecular and RNA Cancer Unit, Kyoto 606-8501, Japan; 2Graduate School of Medicine, Kyoto University, Kyoto 606-8501, Japan; 3Department of Human Genetics, National Institute of Health Doutor Ricardo Jorge, 1649-016 Lisbon, Portugal; 4BioISI–Biosystems & Integrative Sciences Institute, Faculty of Sciences, University of Lisbon, 1749-016 Lisbon, Portugal; 5Laboratory of Cancer cell Biology, Graduate School of Biostudies, Kyoto University, 606-8501 Kyoto, Japan

**Keywords:** Δ160p53, p53, translation, cancer, p53 isoform, p53 mRNA

## Abstract

*p53* is the most commonly mutated gene in human cancers. Two fundamental reasons for this are its long protein isoforms protect from cancer, while its shorter C-terminal isoforms can support cancer and metastasis. Previously, we have shown that the Δ160p53 protein isoform enhances survival and the invasive character of cancer cells. Here, we identified a translation initiation site nine codons downstream of codon 160—the known initiation codon for the translation of Δ160p53—that is recognized by the translation machinery. When translation failed to initiate from AUG160 due to mutation, it initiated from AUG169 instead, producing similar levels of a similar protein, Δ169p53, which promoted cell survival as efficiently as Δ160p53 following DNA damage. Interestingly, almost all mammalian species with an orthologue to human AUG160 also possess one for AUG169, while none of the non-mammalian species lacking AUG160 have AUG169, even if that region of the *p53* gene is well conserved. In view of our findings, we do not believe that Δ169p53 acts as a different p53 protein isoform; instead, we propose that the double translation initiation site strengthens the translation of these products with a critical role in cell homeostasis. Future studies will help verify if this is a more general mechanism for the expression of essential proteins in mammals.

## 1. Introduction

*p53* is a gene well conserved among species, from placozoans to humans [1]. It plays a central role in animals with true tissues (Histozoa) by maintaining the balance between cell growth/differentiation and transformation, and genetic variation and genomic catastrophe. Without *p53,* animals may develop cancer [2], and with too much, they may age prematurely [3]. The multiple and complex tasks of *p53* are tissue-specific and shared between several RNA and protein products [4]. While longer products are well known for activating cell-cycle arrest, senescence and apoptosis, which counter tumor development [5,6,7], shorter C-terminal variants such as Δ160p53, Δ133p53 and Δ133p53β have been reported to induce cancer, invasion and metastasis [8,9]. Previously, we have shown that mutations enriched in cancer activate the translation of Δ160p53 from codon 160, which enhanced cell fitness and invasiveness [8,10]. Here, we identify a second active translation initiation site (TIS) a few nucleotides downstream of codon 160, and we investigate its effect on Δ160p53 expression and function. Understanding the mechanisms of Δ160p53 translation may help us devise a new strategy to target p53 in cancer.

## 2. Results

### 2.1. Toeprinting Identifies Translation Initiation Site Downstream of Codon 160

Since we first identified Δ160p53 as a pro-oncogenic protein in 2013 [8,10], we have been interested in understanding the regulation of this molecule, especially at the translation level, which more clearly separates it from all the other p53 isoforms that can be produced from the same mRNA transcripts. If we could understand the unique regulation of this isoform then we might be able to target it specifically without affecting full-length (FL) p53′s function and that could finally make it possible to target p53 for cancer therapy. One of the common methods used to characterize the mechanisms for the regulation of translation initiation is toeprinting because it measures precisely the position of ribosomes (or ribosomal subunits) on mRNAs and can be used to identify (or confirm) TIS when it is performed in the presence of cycloheximide (CHX), an elongation inhibitor. Toeprinting was performed using in vitro transcribed wild-type (wt) FL *p53* mRNA and two different primers: one well within the coding region of Δ160p53, but still within range of TIS160 (primer 1), and another in the close vicinity of codon 160 (c.160) that also covers Δ160p53′s 5′-untranslated region (5′-UTR) (primer 2). While toeprinting with primer 1 showed no fragments (no peaks in the graph), which means the reverse transcriptase could extend beyond the length limitation of the capillary (around 700 nucleotides) without being stopped by any strong RNA structure or RNA-binding protein (Figure 1A), many were seen with primer 2 (Figure 1B), suggesting that the 5′-UTR possesses strong secondary structures that prevent primer extension. This is a known feature of 5′-UTRs [11]. These structures are usually unwound by the ribosomes and were, with one exception, no longer observed in the presence of ribosomes from rabbit reticulocyte lysates (RRL) (Figure 1C,D). Surprisingly, with RRL and CHX, the reverse transcriptase stopped at c.164 with either primer, which would correspond to the leading edge of 80S ribosomes initiating translation in c.169 and not c.160 [12]. Interestingly though, c.169 is also an AUG and thus a potential TIS. So, we next investigated if c.169 is used as a start site for translation in cells.

### 2.2. Δ169. p53 Is a Newly Identified Translation Product of Full-Length p53 mRNA

We immediately noticed two close bands around 32 KDa (the size of Δ160p53) in several cell lines such as glioblastoma cell line LN229 and A549 lung adenocarcinoma cells (Figure 2A,B). By transfecting a Δ169p53 plasmid made with *p53* codons 169 to 393 into *p53*-null H1299 cells, we saw that the band below Δ160p53 in A549 cells matched the size of Δ169p53 (Figure 2B). H1299 cells expressing FL p53 constructs also exhibited the double band, which became much clearer with cancer-specific mutations R175H and R248Q (Figure 2C). This goes in line with our previous finding showing that common gain-of-function mutations in cancer activate the translation of Δ160p53 and other short p53 isoforms [8]. Interestingly, the new band was induced by DNA damage (treatment with etoposide (Eto)) in LN299 cells (Figure 2D) and endoplasmic reticulum (ER) stress (treatments with thapsigargin (Th) and tunicamycin (Tu)) in wt *p53* breast adenocarcinoma cells MCF7 (Figure 2E), which suggests this protein may play a role in stress response. Finally, we wanted to confirm that this band expressed from FL *p53* transcripts, observed below Δ160p53 and with the size of Δ169p53, was really Δ169p53. So, we created a series of frameshift and TIS mutations. First, to verify that the band of interest was a translation product and not a product of protein degradation or cleavage, we deleted one nucleotide in codon 157 in order to create a frameshift between the sequences upstream and downstream of c.157. We also added an HA-tag to the C-terminus of FL p53 in frame with AUG160 and AUG169, but out of frame—because of the frameshift mutation in c.157—with all other TIS (i.e., TIS 1, 40 and 133). WB against HA revealed no bands corresponding to FL p53, Δ40p53 or Δ133p53, as expected, but presented the double band corresponding to Δ160p53 and Δ169p53, proving that these are products of translation from TIS160 and TIS169, respectively, and not cleavage products of FL p53 or other larger isoforms as those would lack the HA-tag and would not be recognized by the antibody (Figure 3A). Furthermore, a mutation in c.160 shut down Δ160p53′s expression, while enhancing Δ169p53′s, which has its TIS coming next in line [11], and a mutation in c.169 erased Δ169p53′s band, leaving no doubts as to the identity of these proteins. The same TIS mutations removed the same bands in a mutant (R248Q) FL *p53* WB, indicating that in this context also, these proteins are translated directly from TIS160 and TIS169 (Figure 3B).

### 2.3. AUG160 and AUG169 May Have Co-Evolved in Mammals

We next questioned if Δ160p53 and Δ169p53, being so similar, would work together or separately, possess similar functions or not. To gain more insight into the co-existence and possible co-evolution of these two p53 forms, we aligned the *p53* mRNA sequences of 37 different mammalian species (Figure 4A). Of note, human c.160 shows complete conservation in all these species, which suggests Δ160p53 may play a key role in mammalian evolution and physiology. AUG169 orthologs (AUG169^orth^), in turn, are present in 32 (86%) of the species tested. This, when compared to the 81% amino acid conservation in the neighboring codons, also shows some evidence of selective pressure for a second TIS. More so when considering that none of the non-mammalian species lacking an orthologous of AUG160 have one for AUG169 either (0%), even if the neighboring amino acids show 62% conservation to the human peptide sequence (Figure 4B). Altogether, these analyses suggest that it might have been advantageous for most mammals to develop a second TIS downstream of TIS160. On the other hand, species without a TIS160^orth^ never had any need to create a start codon downstream.

### 2.4. TIS160 Is Redundant during Stress Response

Stress of the endoplasmic reticulum (ER) activated the expression of Δ160p53 and Δ169p53 (Figure 2E and Figure 5A). Since HDM2 is a strong regulator of *p53* and is often inactivated during stress [7,13], we first tested if our results could be explained by an alleviation in HDM2-mediated degradation of Δ160p53 and Δ169p53. However, neither HDM2, nor an inhibitor of HDM2, Nutlin-3, had the expected effect on the levels of these proteins (Figure 5B,C, respectively). Therefore, we further investigated the translation dynamics of the two isoforms. Next, we used a dual luciferase reporter system [14] for quick evaluation of translation from one, the other or both (or none) TIS under different cell conditions. Since the wild-type *p53* sequence produced very low signals in this system, we added the cancer mutation R248Q, which enhances expression (Figure 2C). Under normal cell growth, in the presence of a rich medium, removing a single TIS led to a large drop in translation (Figure 5D). Conversely, under mild stress, each TIS alone was sufficient for high expression (Figure 5E). These results are coherent in light of Δ160p53′s pro-survival capacity [8], which would not be crucial under normal growth conditions but would have to be ensured under stress to safeguard the life of the cell. Of course, this will only be true if both protein isoforms, Δ160p53 and Δ169p53, possess similar survival abilities, which we next verified.

### 2.5. Δ160p53 and Δ169p53 Have Similar Survival Capacities

To gain a better idea of the role of TIS169 in cell physiology we compared Δ160p53 and Δ169p53 in their capabilities (or lack of) to affect cell survival under conditions of mild stress (low concentration of etoposide, a DNA damaging agent, or exposure to hydrogen peroxide (H_2_O_2_), which causes oxidative damage) since Δ160p53 was previously shown to promote resistance to mild stress though it failed to rescue cells with severe damage [8]. Etoposide induced cell death over a period of 21 h (27%), but this was greatly minimized (down to 11% and 14%) by either isoform, as both displayed very similar pro-survival qualities (Figure 6A). H_2_O_2_ treatment on the other hand was equally toxic after 24 h, but its effect could not be counteracted by the isoforms (Figure 6B). Importantly, however, in both instances, DNA damage or oxidative damage, Δ160p53 and Δ169p53 proteins led to exactly the same phenotype, supporting our model in which both protein forms play the same role in the cell and function together as the result of a robust bipartite translation initiation process.

## 3. Discussion

Based on our results we propose a model in which, during stress conditions, either AUG, 160 or 169, is sufficient for translation and phenotype manifestation via Δ160p53 and Δ169p53′s functions; though under normal conditions these TIS may be too weak to provide additional survival advantages to the cell, which is desirable (Figure 6C). This double-TIS system seems to have appeared early in mammals, together with Δ160p53, which suggests it is a key regulatory feature of Δ160p53; i.e., for most species, Δ160p53 might not be effective enough or advantageous enough without it. During stress or mutation, translation from TIS169 can be almost as efficient as from TIS160 (Figure 2 and Figure 5) and may play an important role in the cell’s adaptation to stress (Figure 6A) or in promoting tumor formation/development, if permanently active. It will be interesting to investigate if other stress-response genes with important roles in cell-fate decisions also possess comparable mechanisms for the regulation of translation, especially in mammals. HDM2, for example, a known proto-oncogene and p53 regulator with a strong influence on cell fitness also harbors a second downstream AUG in codon 7. In fact, Bazykin et al. observed that downstream AUGs among the first 30 codons are under selection as alternative TIS in eukaryotes, and are particularly common in transcription factors [15]. The authors proposed that these might contribute to increased efficiency of translation and/or for translation of different N-terminal protein variants. Here, our research validates the first hypothesis for the case of TIS169, though we have previously shown that it is the second hypothesis that applies to TIS40 in *p53* [16]. Future research might be able to determine if the exact distance and/or sequence from upstream to downstream AUG is sufficient to predict the role of alternative AUGs in different genes. Our findings also help elucidate the mechanisms of regulation of an essential p53 isoform with possible roles in carcinogenesis and may contribute to a new therapeutic strategy in the future.

## 4. Materials and Methods

### 4.1. Cellular Assays and Reagents

All cell lines were acquired from the American Type Culture Collection (ATCC). Cells were frequently tested for mycoplasma and other contaminations. All the cell lines were maintained at 37 °C and 5% CO_2_ in Dulbecco’s Modified Eagle Media (DMEM) or Roswell Park Memorial Institute (RPMI) media with 10% fetal bovine serum, 5 mM L-glutamine and Pen/Strep 100X solution diluted 1:100. For Western blot analyses (WB) thapsigargin (Sigma; 0.1 μM), tunicamycin (Sigma; 8.5 μM), Nutlin-3 (Calbiochem; 5 μM) and etoposide (Sigma; 5 μM) treatments were for 16 h or as indicated. For WB, cells were lysed in 1.5X SDS sample buffer (62.5 mM Tris-HCl (pH 6.8), 12% glycerol, 2% SDS, 0.004% Bromo Phenol Blue, and 10% 2-mercaptoethanol). After sonication, proteins were separated by SDS-PAGE on 12% gels and transferred to PVDF membranes (ClearTrans® PVDF Membrane (Fujifilm Wako Pure Chemical Corporation, Osaka, Japan)). After blocking for 1 h with blocking buffer (1X TBST with 5% *w/v* non-fat dry milk), membranes were blotted with antibodies diluted in blocking buffer, followed by incubation with secondary antibodies. Membranes were then soaked in Novex ECL (Thermo Fisher Scientific, Waltham, MA, USA) or SuperSignal West Pico PLUS (Thermo Fisher Scientific), and signals were captured with Fujifilm LAS-3000 Imager using different exposure. Primary antibodies (and dilutions) used were rabbit CM1 (1:8000) [17] and mouse Bp53.10 (1:1000) for p53 isoforms, mouse anti-Flag (Sigma M2; 1:2000), rat anti-Flag (BioLegend L5; 1:500), rat anti-HA (Roche 3F10, 1:500), mouse anti-Lamin B1 (Santa Cruz Biotechnology A-11, 1:500), and mouse anti-vinculin (Santa Cruz Biotechnology H-10, 1:500). Secondary antibodies used were anti-mouse IgG HRP-linked antibody (Cell Signaling Technology, Danvers, MA, USA, 1:3000), anti-rabbit IgG HRP-linked antibody (Cell Signaling Technology, 1:3000), and anti-rat IgG HRP-linked antibody (Cell Signaling Technology, 1:3000). All p53 constructs were cloned into pcDNA3.1; luciferase reporter constructs were cloned into psiRF. MTT survival assay was performed according to the manufacturer’s protocol (MTT Cell Proliferation Assay Kit (Cayman Chemical, Ann Arbor, MI, USA)) using 4000 and 5000 cells treated or not with etoposide (Sigma; 20 μM) for 21 h or hydrogen peroxide (Santoku Chemical Industries, Tokyo, Japan; 250 μM) for 24 h. Calibration curves were also performed (2000 to 5000 cells).

### 4.2. Sequence Analyses

Sequence alignment of *p53* from different species was performed using MAFFT [18] server within Jalview software. Sequences used are the following:*Alligator sinensis* (Chinese alligator) XM_006038654.3*Bos taurus* (cattle) NM_174201.2*Callithrix jacchus* (marmoset) XM_002747948.4*Callorhinchus milii* (elephant shark) JN794073.1*Camelus bactrianus* (camel) XM_010965924.1*Canis lupus familiaris* (dog) NM_001003210.1*Carlito syrichta* (tarsier) XM_008062341.2*Catharus ustulatus* (thrush) XM_033084255.1*Danio rerio* (zebrafish) NM_001271820.1*Dasypus novemcinctus* (armadillo) XM_012529094.2*Dipodomys ordii* (kangaroo rat) XM_013013490.1*Echinops telfairi* (lesser tenrec) XM_013007322.2*Enhydra lutris kenyoni* (sea otter) XM_022524640.1*Equus caballus* (horse) XM_023651624.1*Erinaceus europaeus* (hedgehog) XM_007523372.2*Felis catus* (cat) NM_001009294.1*Gallus gallus* (chicken) NM_205264.1*Gorilla gorilla* (gorilla) XM_004058511.3*Homo sapiens* (human) NM_001126112.3*lctidomys tridecemlineatus* (squirrel) XM_005332819.3*Latimeria chalumnae* (coelacanth) XM_005999799.2*Loxodonta africana* (elephant) XM_010596586.2*Macaca mulatta* (Rhesus monkey) NM_001047151.2*Microcebus murinus* (mouse lemur) XM_012776058.2*Mus musculus* (mouse) NM_011640.3*Myotis lucifugus* (bat) XM_006102578.3*Odobenus rosmarus* (walrus) XM_004398491.1*Ornithorhynchus anatinus* (platypus) ENSOANT00000053152.1*Oryctolagus cuniculus* (rabbit) NM_001082404.1*Oryzias latipes* (medaka) NM_001104742.1*Otolemur garnettii* (galago) XM_012806041.2*Pan troglodytes* (chimpanzee) XM_001172077.5*Panthera pardus* (leopard) XM_019413568.1*Pelodiscus sinensis* (turtle) XM_006112136.3*Peromyscus leucopus* (white-footed mouse) XM_028869449.1*Phascolarctos cinereus* (koala) XM_020966468.1*Podarcis muralis* (lizard) XM_028752002.1*Pongo abelii* (orangutan) XM_002826974.4*Puma concolor* (puma) XM_025920288.1*Rattus norvegicus* (rat) NM_030989.3*Sarcophilus harrisii* (Tasmanian devil) XM_031965445.1*Sorex araneu* (shrew) XM_004604858.1*Sus scrota* (pig) NM_213824.3*Trichechus manatus* (manatee) XM_004376021.2*Tursiops truncatus* (dolphin) XM_019944223.2*Ursus arctos horribilis* (grizzly) XM_026520889.1*Xenopus tropicalis* (frog) NM_001001903.1

### 4.3. Toeprinting

Toeprinting was adapted from a previously described protocol [19]. Briefly, 500 ng of in vitro-transcribed (mMESSAGE mMACHINE T7 Ultra kit; Ambion, Austin, TX, USA) full-length *p53* mRNA was combined with 20 pmol of 6-FAM 5′-labelled primer 1 (5′-ATGTAGTTGTAGTGGATGGTGG-3′) or primer 2 (5′-GCCTGGGCATCCTTGAGTTCC-3′) in 50 mM Tris-HCl, pH 7.5, and heated to 68 °C for 2 min and cooled to 37 °C for 8 min, and immediately added to the translational mixture, comprising 20 μM amino acid mixture minus methionine, 20 μM amino acid mixture minus leucine (Promega, Madison, WI, USA), 1.5 U/μL ribonuclease inhibitor (RNaseOUT) plus or minus rabbit reticulocyte lysate (50%), and 500 μg/mL CHX (Sigma). The reactions were incubated at 30 °C for 20 min. Primer extension was performed using SuperScript III Reverse Transcriptase as described by the manufacturer (Invitrogen). Primer extension products were then purified using QIAquick PCR Purification Kit (Qiagen, Hilden, Germany) and run on an Avant 3100 DNA sequencer (ABI) capillary electrophoresis along with a size standard (GS600Liz; Applied Biosystems, Waltham, MA, USA) or a sequencing ladder generated using the same primers, but unlabelled, and analyzed with Peak Scanner 2 software.

## Figures and Tables

**Figure 1 ijms-23-15844-f001:**
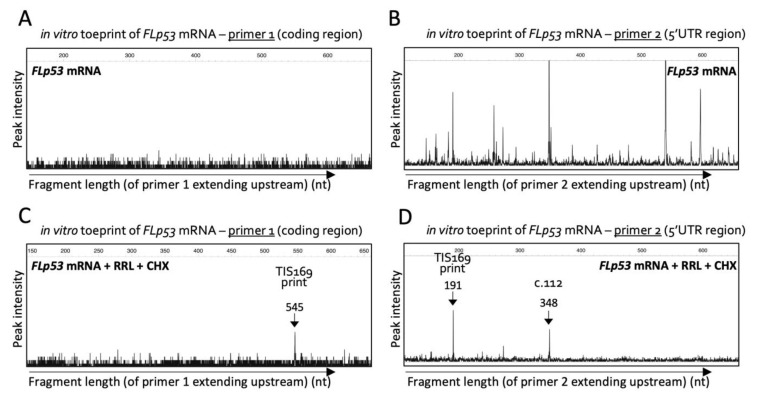
Ribosomal toeprints at AUG169 of in vitro transcribed FL *p53* mRNA observed in the presence of rabbit reticulocyte lysates (RRL) and cycloheximide (CHX).

**Figure 2 ijms-23-15844-f002:**
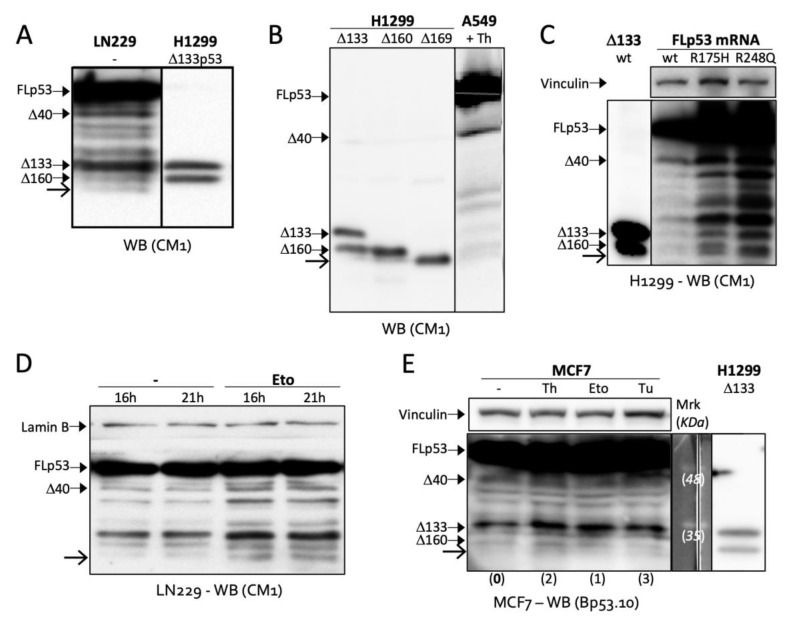
Stress-responsive band (indicated with open arrow) is observed under Δ160p53 band in Western blot analyses (WB) of LN229 (**A**,**D**), A549 (**B**) and MCF7 (**E**) cells endogenously expressing p53 and H1299 (**C**) cells expressing the indicated exogenous constructs. A549 (**B**), LN229 (**D**) and MCF7 (**E**) cells were treated with thapsigargin (Th, 16 h), etoposide (Eto), tunicamycin (Tu, 16 h) or DMSO (−, 16 h) as indicated. Shown are representative data of at least three independent experiments and between brackets (**E**) relative quantifications of the band indicated with the open arrow. CM1, polyclonal anti-p53 antibody; Bp53.10, monoclonal antibody against the C-terminus (aa 374–378) of p53.

**Figure 3 ijms-23-15844-f003:**
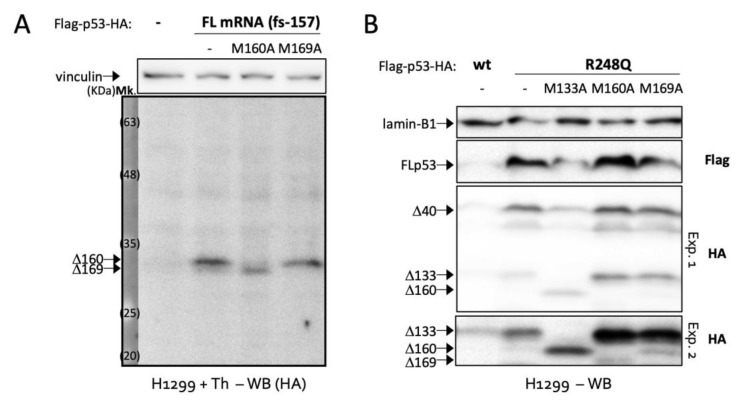
Full-length *p53* mRNA expresses Δ169p53 isoform upon stress or mutation: (**A**) Western blot analyses (WB) of H1299 cells expressing or not full-length (FL) *p53* mRNA, as indicated, and treated with Th (16 h). fs indicates a frameshift mutation of one nucleotide at codon 157 and M160A and M169A indicate mutations in codons 160 and 169, respectively. (**B**) WB of H1299 cells expressing wild-type (wt) or mutant R248Q p53 mRNAs with or without mutated translation start sites (TIS) for Δ133p53, Δ160p53 or Δ169p53 (–, M133A, M160A or M169A). Flag, WB against Flag tag; HA, WB against HA tag. Shown are representative data of at least three independent experiments.

**Figure 4 ijms-23-15844-f004:**
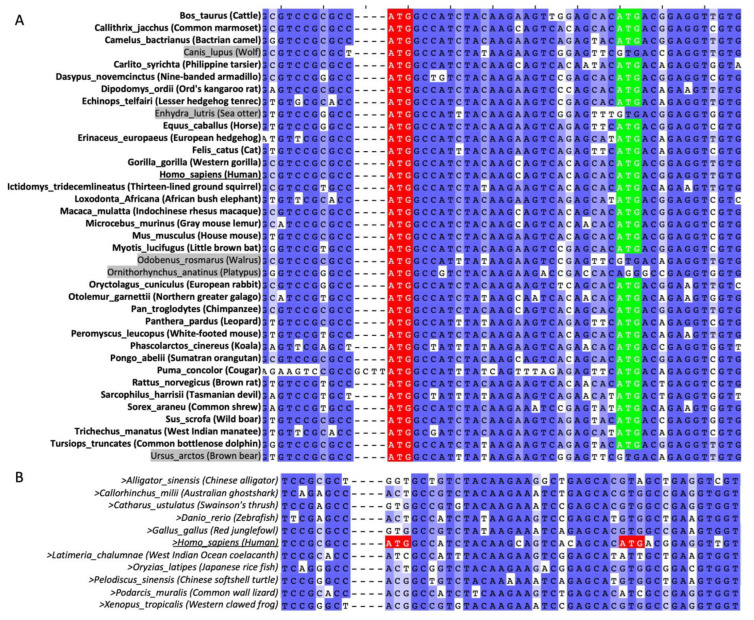
AUG160^orth^ and AUG169^orth^ in different chordate species. Sequence alignment of regions surrounding internal translation initiation sites (TIS) in *p53* from 37 different mammalian species (**A**) or 11 different chordate species (**B**) using MAFFT server within Jalview. In (**A**) AUG orthologs for human AUG160 are shown in red and AUG orthologs for human AUG169 are shown in green.

**Figure 5 ijms-23-15844-f005:**
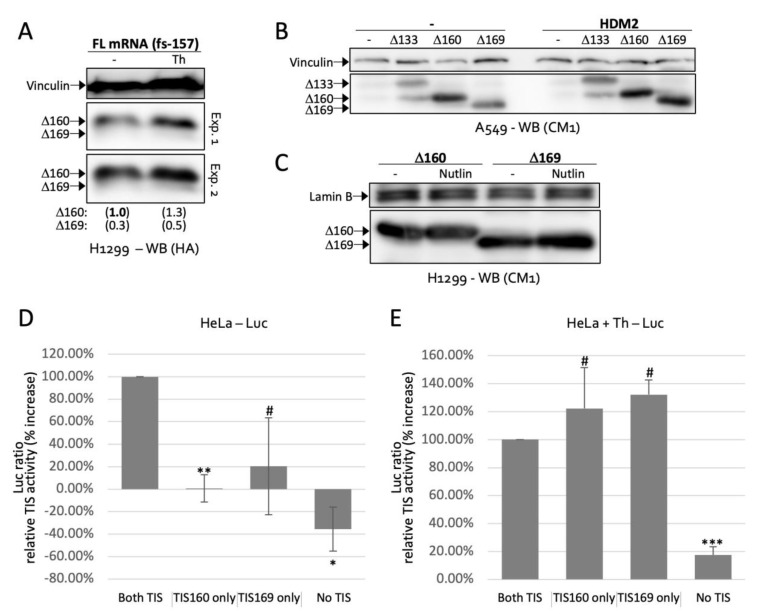
Stress enhances translation from individual translation initiation sites at codons 160 (TIS160) and 169 (TIS169): (**A**–**C**) Western blot analyses (WB) of H1299 or A549 cells expressing the indicated constructs and treated with DMSO (−, 16 h), thapsigargin (Th, 16 h) or Nutlin-3 (Nutlin, 16 h). Values in brackets are quantifications of Δ160p53 and Δ169p53 presented as fold-change relative to Δ160 + DMSO levels and normalized to vinculin (**A**). Exp. 1 and 2 indicate different exposure times (**A**). HA, anti-HA tag antibody; CM1, polyclonal anti-p53 antibody. (**D**,**E**) Luminescence readings (normalized % increase in Firefly Luciferase/Renilla Luciferase ratio over empty control) of HeLa cells expressing dual-luciferase mRNAs containing the first 432 nucleotides of Δ160p53 and no mutation (Both TIS), a mutation in TIS169 (TIS160 only), a mutation in TIS160 (TIS169 only) or both of these mutations (No TIS) and treated with DMSO (**D**) or Th (**E**). Shown are averages ± s.d. of 4 experiments (# *p* > 0.05, * *p* < 0.05, *** p* < 0.01 and **** p* < 0.005 compared to “both TIS” samples).

**Figure 6 ijms-23-15844-f006:**
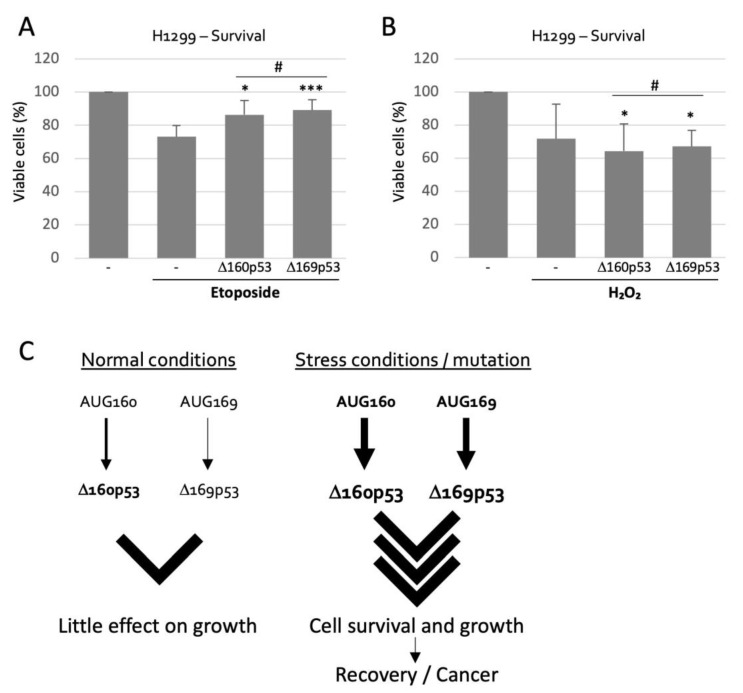
Δ160p53 and Δ169p53 proteins show identical functions: (**A**,**B**) MTT assay (proliferation/survival) of H1299 cells expressing or not Δ160p53 or Δ169p53 and treated or not with DNA damaging agent etoposide (21 h) (**A**) or hydrogen peroxide (H_2_O_2_, 24 h) (**B**), as indicated. Shown are averages ± s.d. of 2–4 experiments (# *p* > 0.05, * *p* < 0.05 and **** p* < 0.005 compared to “Etoposide/− (no *p53* expression)” sample (A) or to “− (no drug and no *p53* expression)” (B) sample or as indicated). (**C**) Proposed model: Under normal cell growth conditions TIS160 and TIS169 show low activity and there is low expression of the corresponding protein isoforms, but under stress conditions or with mutation each of these individual TIS becomes more active and initiate translation of the two protein forms, Δ160p53 and Δ169p53, which have equal capacity to induce cell growth and survival, and which could lead to recovery from stress or, in the case of continued activation, cancer formation or cancer development.

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
