# Peer review of "Conserved Double Translation Initiation Site for Δ160p53 Protein Hints at Isoform’s Key Role in Mammalian Physiology"

_ijms, 2022, doi:10.3390/ijms232415844_

Round 1
Reviewer 1 Report
In this work ``Conserved double translation initiation site for Δ160p53 protein hints at isoform’s key role in mammalian physiology´´ Lopez-Iniesta et al have identified a new transcriptional initiation site (TIS) at codon 169 for a p53 isoform, which previously have been shown by the same group to play an important role in enhancing the survival of cancer cells. The in silico analysis performed by the authors supports a scenario that these two TIS sites AUG 160 and Aug169 of the ∆ 160p53 and ∆169p53 respectively have co-evolved within the mammalian species indicating an advantageous role for the second TIS 169 in supporting the upstream TIS160. Under normal culture condition, both TIS are important for the generation of the ∆ 160p53 transcript but under stress condition each TIS alone was enough for full expression of the isoform. Under mild stress, ∆160p53 as well as ∆169p53 lead to enhanced cell survival. Authors finally propose a model, which suggests that previously described TIS 160 and the newly identified TIS 169 only become active under some stress or oncogenic condition. ∆160p53 and ∆169p53 isoform are equally potent to induce survival under stress, which might lead to cancer development.
Overall, it is an interesting work focusing on role of p53 isomers in oncogenesis. The authors could convincingly established the functionality of the new TIS169, the redundancy of both TIS under stress condition and the pro-survival role of both protein isoforms ∆160p53 and ∆169p53 under mild stress.
Comments:
1. Authors might consider showing the status of endogenous ∆160 p53 and ∆169p53 transcript and or protein in a larger panel of cancer cells carrying endogenous mutated and WT p53, in steady state and under cellular stress.
2. Authors have not addressed the regulation of the ∆160 p53 and ∆169p53 isomers by MDM2. Often mutant p53 proteins escape MDM2 mediated regulation. It will be informative to see if the isomers are regulated by MDM2. This can be shown in vitro by assessing the isomers at protein level upon treatment with Nutlin-3, an inhibitor of MDM2 .
3. Eventually as an outlook, to support the model on possible role of the isomers in oncogenesis, their status should be investigated in tumor samples and the association with the clinical outcome be explored. However, this is probably beyond the scope of the current study.
Author Response
Dear Editors at IJMS and reviewers,
We would like to thank the comments from the reviewers, they really helped improve our manuscript. We have worked hard and performed many experiments and we think the reviewers will be pleased with all the effort and the many new results. Please see the new version of the manuscript and read below our point by point response to the reviewer’s comments:
REVIEWER 1
In this work ``Conserved double translation initiation site for Δ160p53 protein hints at isoform’s key role in mammalian physiology´´ Lopez-Iniesta et al have identified a new transcriptional initiation site (TIS) at codon 169 for a p53 isoform, which previously have been shown by the same group to play an important role in enhancing the survival of cancer cells. The in silico analysis performed by the authors supports a scenario that these two TIS sites AUG 160 and Aug169 of the ∆ 160p53 and ∆169p53 respectively have co-evolved within the mammalian species indicating an advantageous role for the second TIS 169 in supporting the upstream TIS160. Under normal culture condition, both TIS are important for the generation of the ∆ 160p53 transcript but under stress condition each TIS alone was enough for full expression of the isoform. Under mild stress, ∆160p53 as well as ∆169p53 lead to enhanced cell survival. Authors finally propose a model, which suggests that previously described TIS 160 and the newly identified TIS 169 only become active under some stress or oncogenic condition. ∆160p53 and ∆169p53 isoform are equally potent to induce survival under stress, which might lead to cancer development.
Overall, it is an interesting work focusing on role of p53 isomers in oncogenesis. The authors could convincingly established the functionality of the new TIS169, the redundancy of both TIS under stress condition and the pro-survival role of both protein isoforms ∆160p53 and ∆169p53 under mild stress.
We are very pleased to know that reviewer 1 finds our work interesting.
Comments:
- Authors might consider showing the status of endogenous ∆160 p53 and ∆169p53 transcript and or protein in a larger panel of cancer cells carrying endogenous mutated and WT p53, in steady state and under cellular stress.
We have performed the experiments recommended by the reviewer. Figure 2 now includes two new blots (2D and 2E) showing the protein levels of D160 and D169 in two different cell lines (including MCF7 with WT p53) expressing endogenous p53 in steady state and under different types of cellular stress and we could see that these isoforms are induced by stress and respond a little differently to different stresses.
- Authors have not addressed the regulation of the ∆160 p53 and ∆169p53 isomers by MDM2. Often mutant p53 proteins escape MDM2 mediated regulation. It will be informative to see if the isomers are regulated by MDM2. This can be shown in vitro by assessing the isomers at protein level upon treatment with Nutlin-3, an inhibitor of MDM2 .
We have now addressed the regulation of D160 and D169 by MDM2 by using Nutlin-3 as suggested by the reviewer and also by co-expressing MDM2. These had little or no effect on isoform levels, so it would seem they also escape MDM2 mediated regulation as predicted by the reviewer. Please see the results in new figures 5B and 5C.
- Eventually as an outlook, to support the model on possible role of the isomers in oncogenesis, their status should be investigated in tumor samples and the association with the clinical outcome be explored. However, this is probably beyond the scope of the current study.
We completely agree with the reviewer that it will be very interesting to investigate the status of the isoforms in tumor samples but also think it is beyond the scope of the current study.
Reviewer 2 Report
As an extension of their previous work on Δ160p53 mutant, the authors of this manuscript reported further findings of AUG169 as a second active translation initiation site downstream of AUG160. Preliminary studies suggest a novel mechanism of double translation initiation sites of AUG160 and AUG169 which promote cell survival under stress conditions. These interesting results would be of great interest to the basic research of p53 as well as cancer therapy targeting p53 mutations. Therefore, this manuscript could be considered for publication after the following revisions.
Translation from either TIS169 or TIS160 was shown in a luciferase assay to increase under the stress condition by thapsigargin treatment. Can the enhanced expression by TIS169 or TIS160 be seen at mRNA or protein level?
The pro-survival effect of Δ160p53 or Δ169p53 was only demonstrated under only genotoxicity-induced condition (etoposide). It would be more convincing to verify this effect under other stress conditions such as oxidative stress (e.g., peroxide), irradiation etc.
In the cell viability assay (MTT), both Δ160p53 and Δ169p53 were shown to promote cell survival upon etoposide-induced cell death. Are there any changes in p53 target genes involved in cell cycle or apoptosis?
Author Response
Dear Editors at IJMS and reviewers,
We would like to thank the comments from the reviewers, they really helped improve our manuscript. We have worked hard and performed many experiments and we think the reviewers will be pleased with all the effort and the many new results. Please see the new version of the manuscript and read below our point by point response to the reviewer’s comments:
REVIEWER 2
As an extension of their previous work on Δ160p53 mutant, the authors of this manuscript reported further findings of AUG169 as a second active translation initiation site downstream of AUG160. Preliminary studies suggest a novel mechanism of double translation initiation sites of AUG160 and AUG169 which promote cell survival under stress conditions. These interesting results would be of great interest to the basic research of p53 as well as cancer therapy targeting p53 mutations. Therefore, this manuscript could be considered for publication after the following revisions.
We are pleased to know that reviewer 2 finds our results to be of great interest to the basic research of p53 as well as cancer therapy and so we have revised the manuscript thoroughly according to the comments of the reviewers.
Translation from either TIS169 or TIS160 was shown in a luciferase assay to increase under the stress condition by thapsigargin treatment. Can the enhanced expression by TIS169 or TIS160 be seen at mRNA or protein level?
Yes, we now have 2 new results confirming this in 2 different cell lines: new figures 2E and 5A.
The pro-survival effect of Δ160p53 or Δ169p53 was only demonstrated under only genotoxicity-induced condition (etoposide). It would be more convincing to verify this effect under other stress conditions such as oxidative stress (e.g., peroxide), irradiation etc.
We now tested oxidative stress as suggested by the reviewer (new figure 6B) and even though there was no increase in proliferation with the expression of the isoforms they both showed exactly the same phenotype, which is the point we want to make and that is more important for our conclusions: that D160p53 and D169p53 proteins work exactly the same way. So we think it’s good to have both examples, one when both are active and one when both are inactive, to make our point more conclusive. We also refer to our previous work where we showed several pro-oncogenic functions of D160p53 (Candeias 2016).
In the cell viability assay (MTT), both Δ160p53 and Δ169p53 were shown to promote cell survival upon etoposide-induced cell death. Are there any changes in p53 target genes involved in cell cycle or apoptosis?
We have performed some of these experiments (see below) but do not wish to include them in this manuscript as we think this manuscript focuses on the (new type of) regulation of D160/D169 at the translational level and not on the mechanisms of action of these isoforms. Also, the results are negative, so the mechanisms are still unknown. This is somewhat expected because D160p53 and D169p53 have lost the transactivation domains of FLp53.

Reviewer 3 Report
In this manuscript, the authors report the detection of the alternative translation initiation codon of p53 isoform, which starts its translation in codon 160 – thus the translation of the new form starts from codon 169. Both forms have oncogenic properties – when overexpressed they promote the survival of cells.
The paper is well-written, as far as I can judge, the conclusions are supported by data. I am not convinced if the paper is of sufficient interest to the general reader of IJMS.
There are some points, which must be explained if the paper is to be published:
1. Much of the inspiration for the experiments was derived from the results of ribosomal toeprinting experiments, however both the method and the results of this experiment are not sufficiently explained. The authors should improve it as this method is not frequently used and is not widely-known.
2. The authors declare that the bands visible on Western blots with lower molecular weight than the full-length p53 are derived from the alternative translation initiation sites of full-length mRNA (Fig 2). They mark Δ40, Δ133, Δ160 and Δ169 forms. However, there are other bands visible between Δ40 and Δ133 forms. Does that mean that the usage of the alternative translation initiation sites is very common in the production of p53? Judging by the intensity of the Western blot bands of the full-length p53 and the ones for the alternative translation it can be concluded that the alternative forms constitute small fraction of p53 protein in cells. Do they really play a role in cell physiology or are they just some background noise of p53 production? Please discuss.
Author Response
Dear Editors at IJMS and reviewers,
We would like to thank the comments from the reviewers, they really helped improve our manuscript. We have worked hard and performed many experiments and we think the reviewers will be pleased with all the effort and the many new results. Please see the new version of the manuscript and read below our point by point response to the reviewer’s comments:
REVIEWER 3
In this manuscript, the authors report the detection of the alternative translation initiation codon of p53 isoform, which starts its translation in codon 160 – thus the translation of the new form starts from codon 169. Both forms have oncogenic properties – when overexpressed they promote the survival of cells.
The paper is well-written, as far as I can judge, the conclusions are supported by data. I am not convinced if the paper is of sufficient interest to the general reader of IJMS.
Obviously on this matter we are more in agreement with reviewers 1 and 2 who think this study is of great interest “to the basic research of p53 as well as cancer therapy targeting p53 mutations”. This is especially true because our paper was submitted to a special issue on p53. We also think that it will be of interest to researchers in other fields of cancer and translation regulation and more, as it presents a new idea/concept in molecular biology, evolution and physiology: double TIS for double security.
There are some points, which must be explained if the paper is to be published:
- Much of the inspiration for the experiments was derived from the results of ribosomal toeprinting experiments, however both the method and the results of this experiment are not sufficiently explained. The authors should improve it as this method is not frequently used and is not widely-known.
We have re-written and greatly extended this section in the manuscript and believe it will now be clear. We also give reference to previous work using this method and to the paper that first described the method.
- The authors declare that the bands visible on Western blots with lower molecular weight than the full-length p53 are derived from the alternative translation initiation sites of full-length mRNA (Fig 2). They mark Δ40, Δ133, Δ160 and Δ169 forms. However, there are other bands visible between Δ40 and Δ133 forms. Does that mean that the usage of the alternative translation initiation sites is very common in the production of p53? Judging by the intensity of the Western blot bands of the full-length p53 and the ones for the alternative translation it can be concluded that the alternative forms constitute small fraction of p53 protein in cells. Do they really play a role in cell physiology or are they just some background noise of p53 production? Please discuss.
We thank the reviewer for the interesting points raised. I will try to explain. Not all the bands lower than FLp53 are alternative translation products. We can tell with the frameshift mutations, AUG mutations and the use of N- and C-terminal tags and different p53 antibodies and mass-spec. One of the missions of our lab is actually to identify all the bands, and we are being quite successful. There are several N-terminal fragments between D40p53 and D133 that result from cleavage of FLp53, and there are C-terminal fragments below D169 and one more alternative translation product that we will be describing soon in a new publication. Some bands are also post-translationally modified isoforms, with ubiquitin, for example. It is now clear that isoforms like D133 and D160/D169 play an important role in the cell (and organism) as shown by knock-down studies and also animal studies [1–3]. The levels are not always low, it’s not uncommon that D133 or D160/D169 levels reach 0.3-0.5 of FLp53 levels (see blots below), which is very significative since D160/D169 binds 3 times more strongly to FLp53 than FLp53 itself (our unpublished data; beyond the scope of this paper).
- Niu, G.; Hellmuth, I.; Flisikowska, T.; Pausch, H.; Rieblinger, B.; Carrapeiro, A.; Schade, B.; Böhm, B.; Kappe, E.; Fischer, K.; et al. Porcine model elucidates function of p53 isoform in carcinogenesis and reveals novel circTP53 RNA. Oncogene 2021, 40, 1896–1908.
- Candeias, M.M.; Hagiwara, M.; Matsuda, M. Cancer‐specific mutations in p53 induce the translation of Δ160p53 promoting tumorigenesis. EMBO Rep. 2016, 17, 1542–1551.
- Fujita, K.; Mondal, A.M.; Horikawa, I.; Nguyen, G.H.; Kumamoto, K.; Sohn, J.J.; Bowman, E.D.; Mathe, E.A.; Schetter, A.J.; Pine, S.R.; et al. P53 Isoforms Δ133P53 and P53Β Are Endogenous Regulators of Replicative Cellular Senescence. Nat. Cell Biol. 2009, 11, 1135–1142.

Round 2
Reviewer 2 Report
The revised manuscript has been significantly improved and is now suitable for publication.